# ClusterPrompt: Cluster Semantic Enhanced Prompt Learning for New Intent Discovery

**Jinggui Liang**
Singapore Management University
`jg.liang.2023@phdcs.smu.edu.sg`

**Lizi Liao**
Singapore Management University
`lzliao@smu.edu.sg`

## Abstract

The discovery of new intent categories from user utterances is a crucial task in expanding agent skills. The key lies in how to efficiently solicit semantic evidence from utterances and properly transfer knowledge from existing intents to new intents. However, previous methods laid too much emphasis on relations among utterances or clusters for transfer learning, while paying less attention to the usage of semantics. As a result, these methods suffer from in-domain over-fitting and often generate meaningless new intent clusters due to data distortion. In this paper, we present a novel approach called Cluster Semantic Enhanced Prompt Learning (CsePL) for discovering new intents. Our method leverages two-level contrastive learning with label semantic alignment to learn meaningful representations of intent clusters. These learned intent representations are then utilized as soft prompt initializations for discriminating new intents, reducing the dominance of existing intents. Extensive experiments conducted on three public datasets demonstrate the superiority of our proposed method. It not only outperforms existing methods but also suggests meaningful intent labels and enables early detection of new intents.

## 1 Introduction

New Intent Discovery (NID) aims to automatically identify novel intent categories that are not defined or observed beforehand. It plays a critical role in task-oriented dialogue systems with the ability to discern newly emerging user preferences (Liao et al., 2023a,b), thereby providing high-quality services (Lin et al., 2020; Zhang et al., 2021c, 2023). Different from the traditional intent classification (E et al., 2019; Chen et al., 2019; Wang et al., 2021), the key challenges of NID lie in how to properly transfer the prior knowledge from exist-

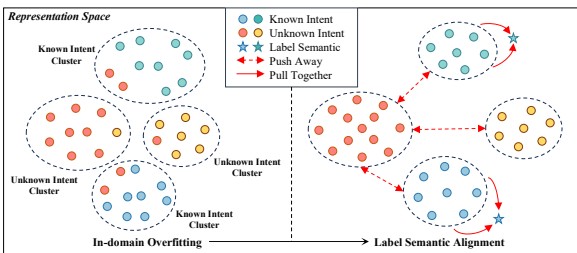

Figure 1: The overview of the in-domain over-fitting problem in NID and our label semantic alignment.

ing intents to discover new intents and efficiently solicit semantic evidence from user utterances.

Existing NID methods can be divided into two categories: unsupervised and semi-supervised. In unsupervised NID, researchers mainly focus on how to extract better utterance features to assist the clustering (Padmasundari and Bangalore, 2018; Shi et al., 2018). But they tend to ignore the prior knowledge contained in the labeled data. Thus, in semi-supervised NID, various methods train representation learning models to facilitate knowledge transfer between labeled and unlabeled data, then perform clustering on utterance representations for NID in a two-stage fashion (Zhang et al., 2023; Zhou et al., 2022). In view of knowledge transfer, there are works that first pre-train an in-domain intent classifier, and then gradually update it with clustered pseudo-labels (Lin et al., 2020; Zhang et al., 2021c). There are also works that formulate contrastive learning objectives to optimize model parameters (Mou et al., 2022b; Zhang et al., 2022).

The essence of these two-stage methods lies in learning discriminative semantic features for utterances via minimizing intra-class variance and maximizing inter-class variance in the first stage. Hence, the similarity or dis-similarity relations between utterances or clusters are emphasized. For instance, pair-wise similarities are used as pseudo supervision in (Lin et al., 2020) and various contrastive

learning objectives naturally enforce such relations (Wei et al., 2022; Mou et al., 2022a). However, the clustering process in the second stage can be easily distorted in favor of labeled data and dominant intent categories, resulting in the in-domain over-fitting problem as shown in Figure 1. Another risk is that such relation distortion would obscure the semantic meaning of intent clusters, leading to less meaningful new intents.

In this paper, we thus propose a **C**luster **s**emantic **e**nhanced **P**rompt **L**earning (CsePL) method for NID as two stages. Specifically, we leverage the semantic knowledge to regulate both of the two stages, which are formulated as Intent Cluster Representation Learning (ICRL), and Prompting for Intent Discrimination (PID). In the ICRL stage, besides using two-level contrastive learning objectives to learn compact and closely connected regions for intents in feature space, we align the intent cluster representations with their corresponding label semantics. It enables the model to learn stable semantic features and semantic-aware intent cluster representations. In the following PID stage, we employ the learned intent cluster representations for soft prompt initializations and integrate them into input utterances to facilitate new intent discrimination. Given that the new inputs encompass the semantics from all intents, the prompting mechanism will encourage the model to focus on matching the utterances with their inherent semantic meaning, thus reducing the dominance of existing intents. We evaluate the proposed CsePL on three widely-used datasets. It outperforms state-of-the-art (SOTA) methods in various aspects.

To summarize, our contributions are three-fold:

- In view of the in-domain over-fitting and meaningless new intent problems, we propose to reiterate the importance of semantic knowledge in utterances and intent clusters for NID.
- We propose two-level contrastive learning objectives with label semantic alignment for learning semantic-aware intent cluster representations and leverage the soft prompting mechanism to enhance the usage of semantic knowledge in intent discrimination.
- Experiments show that CsePL not only gains significant improvements over SOTA methods, but also suggests meaningful intent labels and enables early detection of new intents.

## 2   Related Work

**New Intent Discovery.**   Identifying new intents are key to adaptable conversational agents for better dialogue state understanding (Zhang et al., 2019; Liao et al., 2021). Previous research on NID can be predominantly categorized into two types: unsupervised and semi-supervised. For the former, early approaches (Cheung and Li, 2012; Li et al., 2013) primarily relied on statistical features of the unlabeled data to cluster similar queries for discovering new user intents. Subsequently, some studies (Xie et al., 2016; Yang et al., 2017; Shi et al., 2018; Hadifar et al., 2019) have endeavored to leverage deep neural networks to learn robust representations conducive to new intent clustering. However, none of these fully leveraged supervised signals, such as existing intent labels.

To address this, recent studies (Lin et al., 2020; Zhang et al., 2021b,c; Mou et al., 2022b; Zhang et al., 2022) have extended NID to a semi-supervised setting to achieve prior knowledge transfer, in which the labeled data is incorporated into the training process to assist new intent clustering. For example, Wei et al. (2022) and Zhang et al. (2023) first pre-trained a backbone model with the supervision of the limited labeled data. Then, they employed the pre-trained backbone to generate pseudo labels for the unlabeled data, directing the model to discern novel intents. Additionally, different from those pseudo-labeling based semi-supervised methods, Mou et al. (2022a) and Zhang et al. (2022) sought to directly optimize utterance representations with the aid of supervised data. They formulated distinct contrastive learning objectives to learn discriminative utterance representations, facilitating the similar utterance clustering and establishing distinct boundaries for new intent clusters.

However, all these methods overemphasize on relations such as similarity or dissimilarity among utterances for better clustering effects, while belittle the usage of semantics inside utterances and intents. Similar to the trivial solution in clustering (Yang et al., 2017; Caron et al., 2018; Ji et al., 2019; Zhang et al., 2021a; Zheng et al., 2023), it brings the problem of in-domain over-fitting and meaningless new intents due to the data distortion. In our work, we enhance model with semantic knowledge and use soft prompts to detect new intents.

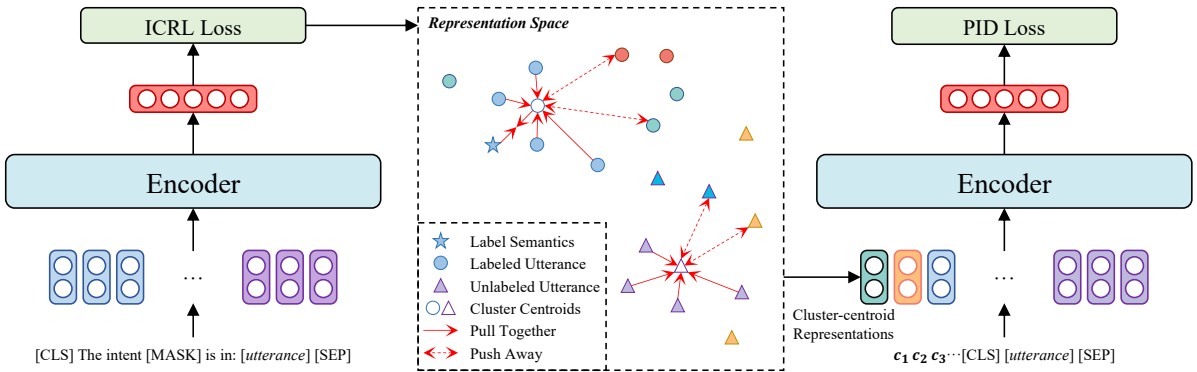

Figure 2: The overall architecture of our proposed CsePL framework for new intent discovery. The left part denotes the ICRL stage and the right part is the PID stage, where [*utterance*] is the original utterance, and $\{c_1, c_2, c_3, ...\}$ are the soft prompts initialized by all the learned intent cluster representations.

**Prompt Learning.** It is a new NLP paradigm of leveraging pre-trained language models (PLMs) (Brown et al., 2020; Sanh et al., 2022; Deng et al., 2023), which reformulates downstream tasks by inserting task-specific instructions into the input to align them with pre-training tasks. Early works (Jiang et al., 2020; Shin et al., 2020; Yuan et al., 2021; Ben-David et al., 2022) mainly utilized discrete hand-crafted or automatically searched prompts to acquire knowledge from PLMs. However, since discrete prompts are hard to optimize, recent works (Li and Liang, 2021; Lester et al., 2021; Liu et al., 2021; Gu et al., 2022; Hou et al., 2022) in prompt learning made efforts to optimize soft prompts in the continuous embedding space. It is more flexible and performs well in various downstream tasks. Here we explore to utilize both discrete prompts and soft prompts to leverage semantic knowledge for NID.

**Contrastive Learning.** Contrastive learning is a popular and effective approach to learn discriminative representations in both computer vision and NLP tasks (Chen et al., 2020; He et al., 2020; Fang and Xie, 2020; Carlsson et al., 2021; Giorgi et al., 2021; Gao et al., 2021; Wu et al., 2022a,b; Ye et al., 2022; Guo et al., 2022). The primary intuition of contrastive learning is to pull together positive pairs in feature space, while push away negative pairs. Motivated by its superior performance, contrastive learning has also been adopted to intent recognition in recent works (Zhang et al., 2021d; Mou et al., 2022a; Wei et al., 2022; Zhang et al., 2022, 2023). We leverage it to help us learn better utterance and intent cluster representations.

## 3 The CsePL Approach

### 3.1 Problem Formulation

Let $\mathcal{C}_k$ and $\mathcal{C}_u$ denote the known intent set and unknown intent set respectively, where $\mathcal{C}_k \cap \mathcal{C}_u = \varnothing$. In general, given a set of labeled data $\mathcal{D}_{known}^{labled} = \{(x_i, y_i)|y_i \in \mathcal{C}_k\}$ and a set of unlabeled data $\mathcal{D}^{unlabeled} = \{x_i|y_i \in \mathcal{C}_k \cup \mathcal{C}_u\}$, the goal of NID is to identify potential unknown intents in $\mathcal{C}_u$ from $\mathcal{D}^{unlabeled}$ and classify the input $x_i$ into its corresponding intent $y_i$, where $y_i \in \mathcal{C}_k \cup \mathcal{C}_u$. Here we focus on the semi-supervised NID setting.

### 3.2 Model Overview

The proposed CsePL model is illustrated in Figure 2, which consists of two stages for discovering new intents. The first stage is Intent Cluster Representation Learning (ICRL) while the second stage is Prompting for Intent Discrimination (PID). We introduce these two stages in the following subsections one by one. In general, the ICRL stage is designed to solicit meaningful intent cluster representations. To achieve this, we reorganize the user utterances using a unified hand-crafted discrete prompt, which are then provided as inputs to a BERT-based backbone for feature extraction. Then the two-level contrastive learning with label semantic alignment objectives are applied to optimize model parameters. Given the learned intent cluster representations, the PID stage targets discriminating intents for all utterances under the all intents-aware situation. Hence, we construct soft prompts with these learned semantic-aware intent cluster representations as initializations and perform further fine-tuning to guide the pre-trained

backbone to discriminate intents. During inference, we send all test utterances to the well-trained PID model to extract utterances' intents-aware representations, and then conduct $K$-means to predict the intent categories for them.

### 3.3 Intent Cluster Representation Learning

Different from previous approaches that emphasis on relations among utterances or clusters, in this stage, we aim to enhance our model with semantic knowledge inside utterances and intents to learn meaningful intent cluster representations.

To achieve this, we first employ a prompt learning method to extract representations for the input utterances in both $\mathcal{D}_{known}^{labeled}$ and $\mathcal{D}^{unlabeled}$. As aforementioned, it is an effective approach to leverage PLMs to extract semantic information inside utterances. Given an input utterance $x_i$, we convert $x_i$ to $x_i^{prompt}$ by inserting a unified hand-crafted discrete prompt into it. The prompted utterance $x_i^{prompt}$ is given as:

$$x_i^{prompt} = \text{[CLS] The intent [MASK] is in: } x_i \text{ [SEP]},$$

where "[CLS] The intent [MASK] is in:" are hand-crafted discrete prompt tokens. We tried different token designs and empirically chose these for best performance. We sent $x_i^{prompt}$ to PLMs, and regard the extracted representation $z_i$ at position "[MASK]" as the representation of input $x_i$.

Then, we conduct two-level contrastive learning with label semantic alignment to optimize the obtained utterance representations for learning meaningful intent cluster representations. In the utterance-level, we conduct both supervised and unsupervised contrastive learning to learn more accurate utterance representations. In the cluster level, we enforce the cluster center to be far away from other cluster utterances and close to its own cluster members. Beyond these, we further use label semantics to regulate the intent cluster center representations via contrastive learning objective.

**Utterance-level Contrastive Loss.** Inspired by the contrastive learning scheme (Khosla et al., 2020) under supervised setting, we optimize the model to bring utterances under the same intent together while disperse utterances from different intents apart. Let $y_i$ denotes the ground-truth intent label for an utterance $x_i$ and $\mathcal{Y}(i)$ denote all

utterances sharing the same intent label as $x_i$, the utterance-level supervised contrastive loss is:

$$\mathcal{L}_{utt}^{cl} = - \sum_{j \in \mathcal{Y}(i)} log \ \frac{exp(\boldsymbol{f}_i \cdot \boldsymbol{f}_j^T / \tau_1)}{\sum_{i \neq l} exp(\boldsymbol{f}_i \cdot \boldsymbol{f}_l^T / \tau_1)}, \ (1)$$

where $\boldsymbol{f}_i = \phi(\boldsymbol{z}_i)$, $\tau_1$ denotes the temperature in contrastive loss, and $\phi$ is a normalizing projection.

Beyond the label supervision, we also conduct unsupervised contrastive learning to help learning utterance representations similar to Equation 1. For the utterance $x_i$, its positive sample is derived from its dropout-augmented view $\widetilde{x}_i$. Meanwhile, the negative samples for $x_i$ are taken from all other utterances and their corresponding augmentations within the same mini-batch.

Moreover, since intent labels are high-quality supervision signals, we also adopt cross-entropy loss for labeled utterances to optimize the model.

**Cluster-level Contrastive Loss.** In the cluster level, we optimize contrastive objectives to pull each cluster representation close to its utterance members while push away from other intent cluster utterances. We conduct $K$-means to obtain intent clusters from $\mathcal{D}^{unlabeled}$ at the beginning of each training epoch. Suppose a cluster $S_i = \{x_1, x_2, \cdots, x_q\}$, we directly define the cluster center $\boldsymbol{c}_i$ for the cluster as:

$$\boldsymbol{c}_i = \frac{1}{|S_i|} \sum_{j=1}^{|S_i|} \boldsymbol{f}_j \ .$$

Therefore, the cluster-level contrastive learning objective is defined as:

$$\mathcal{L}_{clu}^{cl} = - \sum_{i=1}^{|S_i|} log \ \frac{exp(\boldsymbol{f}_i \cdot \boldsymbol{c}_i^T / \tau_2)}{\sum_{i \neq j} exp(\boldsymbol{f}_i \cdot \boldsymbol{c}_j^T / \tau_2)}, \quad (2)$$

where $\boldsymbol{c}_i$ is the corresponding intent cluster center representation of the utterance $x_i$, while $\boldsymbol{c}_j$ represents any intent cluster center that is not of $x_i$ and $\tau_2$ denotes the temperature in this contrastive loss.

**Semantic Alignment.** For the labeled training data, we investigate to solicit semantic information from both utterances and intent labels and optimize the model with label semantic alignment. The intuition behind this is that, the label semantic features are more stable in the feature space and will not be influenced by the distribution of the training data.

Aligning the known intent cluster representations to such stable label semantic features can protect the model from in-domain over-fitting problem and enable the model to give suggestions about intent labels. To achieve this, we also adopt a contrastive learning objective for label semantic alignment.

Specifically, for each known intent $y_i \in \mathcal{C}_k$, we first use the BERT-based model to get embeddings for tokens in intent label $y_i$. As each intent label may contain multiple tokens, we directly apply mean pooling operation to obtain intent label representation as:

$$\hat{\boldsymbol{c}}_i = \frac{1}{|y_i|} \sum_{j=1}^{|y_i|} BertEmb\left(t_j\right), \quad (3)$$

where $|y_i|$ is the number of tokens in the intent label $y_i$ and *BertEmb*$(\cdot)$ is the embedding projection in BERT-based model. $t_j$ denotes the $j$-th token in the $y_i$. Then, the label semantic alignment loss is defined as follows:

$$\mathcal{L}_{align}^{cl} = -\sum_{i=1}^{|\mathcal{C}_k|} log \, \frac{exp(\boldsymbol{c}_i \cdot \hat{\boldsymbol{c}}_i^T / \tau_3)}{\sum_{i \neq j} exp(\boldsymbol{c}_i \cdot \hat{\boldsymbol{c}}_j^T / \tau_3)}, \quad (4)$$

where $|\mathcal{C}_k|$ is the number of known intents and $\tau_3$ denotes the temperature. Note that cluster indices generated by *K*-means are permuted randomly in different training epoch. To tackle this pseudo label inconsistency problem and provide high-quality supervised signals, we also conduct cluster alignment following the work of Zhang et al. (2021c).

As a result, the overall loss function in the ICRL stage can be formulated as follows:

$$\mathcal{L}_{ICRL} = \mathcal{L}_{ce} + \alpha \mathcal{L}_{utt}^{cl} + \beta \mathcal{L}_{utt}^{ucl} + \lambda \mathcal{L}_{clu}^{cl} + \eta \mathcal{L}_{align}^{cl},$$

where $\mathcal{L}_{ce}$ is the cross-entropy loss. $\{\alpha, \beta, \lambda, \eta\}$ represent hyper-parameters that modulate the respective contributions of distinct losses.

### 3.4 Prompting for Intent Discrimination

In the former stage, we have trained the model with different contrastive learning objectives to learn meaningful intent cluster representations. In the PID stage, we apply soft prompt learning to exploit these learned representations for efficient intent discrimination. It is an effective and flexible approach to leverage the semantic knowledge of PLMs and demonstrates strong performance across

a range of downstream tasks. Specifically, we utilize the learned intent cluster representations as soft prompt initializations. Given an input utterance $x_i$, we first use the pre-trained backbone to convert $x_i$ into a sequence of token embeddings $E_i = \{\boldsymbol{e}_{[CLS]}, \boldsymbol{e}_1, \cdots, \boldsymbol{e}_{[SEP]}\}$. Then, we insert a sequence of soft prompt vectors to $E_i$ to construct the formally input as:

$$E_{prompt} = \{\boldsymbol{c}_1, \cdots, \boldsymbol{e}_{[CLS]}, \boldsymbol{e}_1, \cdots, \boldsymbol{e}_{[SEP]}\},$$

where $\boldsymbol{c}_i$ is the intent cluster representation learned in the ICRL stage. The prompted input $E_{prompt}$ is send to the pre-trained backbone to extract representation for $x_i$. In this stage, we regard the extracted representation at position "[CLS]" after normalizing projection as the utterance representation $\boldsymbol{h}_i$. It's noteworthy that each derived intent cluster representation acts as an independent soft prompt token in the prompted input, which reveals all the intent candidate semantics to the model. To update the model in this stage, following the work of Zhang et al. (2022), we optimize a contrastive learning objective, which mines neighboring utterance representations and pulls them together and pushes away distant ones in the feature space. The contrastive learning loss is calculated as:

$$\mathcal{L}_{PID} = -\sum_{p \in \mathcal{N}(i)} log \, \frac{exp(\boldsymbol{h}_i \cdot \boldsymbol{h}_p^T)/\tau_4)}{\sum_{i \neq j} exp(\boldsymbol{h}_i \cdot \boldsymbol{h}_j^T)/\tau_4)},$$

where $\mathcal{N}(i)$ is the neighbor utterance set of $x_i$ and $\tau_4$ denotes the temperature. Similar to Zhang et al. (2022), we select the most similar 50 utterances to $x_i$ in the feature space as its neighbors. Here $n$ is the mini-batch size and each utterance is accompanied with an augmented version. During training, we update the neighbor utterance set $\mathcal{N}(i)$ every few epochs to guide the model to form clear cluster boundaries for new intent discovery.

## 4 Experiments

### 4.1 Datasets

We conduct experiments to evaluate the performance of our CsePL on three widely-used NID datasets. **Banking77** (Casanueva et al., 2020) is a fine-grained banking domain dataset and comprises 13,083 customer service queries over 77 intents. **Clinc150** (Larson et al., 2019) is a multi-domain

| Known Intent Rate | Methods | Banking77 | | | Clinc150 | | | StackOverflow | | |
|---|---|---|---|---|---|---|---|---|---|---|
| | | NMI | ARI | ACC | NMI | ARI | ACC | NMI | ARI | ACC |
| 25% | DTC | 55.59 | 19.09 | 31.75 | 79.35 | 41.92 | 56.90 | 29.96 | 17.51 | 29.54 |
| | CDAC+ | 66.39 | 33.74 | 48.00 | 84.68 | 50.02 | 66.24 | 46.16 | 30.99 | 51.61 |
| | DeepAligned | 70.50 | 37.62 | 49.08 | 88.97 | 64.63 | 74.07 | 50.86 | 37.96 | 54.50 |
| | DCSC | 78.18 | 49.75 | 60.15 | 91.70 | 72.68 | 79.89 | - | - | - |
| | MTP-CLNN | 80.04 | 52.91 | 65.06 | 93.17 | 76.20 | 83.26 | 73.35 | 54.80 | 74.70 |
| | USNID | 81.94 | 56.53 | 65.85 | **94.17** | 77.95 | 83.12 | **74.91** | **65.45** | 75.76 |
| | CsePL | **83.32**$^\dagger$ | **60.36**$^\dagger$ | **71.06**$^\dagger$ | 94.07$^-$ | **79.65**$^\dagger$ | **86.16**$^\dagger$ | 74.88$^-$ | 64.92$^-$ | **79.47**$^-$ |
| 50% | DTC | 69.46 | 37.05 | 49.85 | 83.01 | 50.44 | 64.39 | 49.80 | 37.38 | 52.92 |
| | CDAC+ | 67.30 | 34.97 | 48.55 | 86.00 | 54.87 | 68.01 | 46.21 | 30.88 | 51.79 |
| | DeepAligned | 76.67 | 47.95 | 59.38 | 91.59 | 72.56 | 80.70 | 68.28 | 57.62 | 74.52 |
| | DCSC | 81.19 | 56.94 | 68.30 | 93.75 | 78.82 | 84.57 | - | - | - |
| | MTP-CLNN | 83.42 | 60.17 | 70.97 | 94.30 | 80.17 | 86.18 | 76.66 | 62.24 | 80.36 |
| | USNID | 85.05 | 63.77 | 73.27 | **95.45** | 82.87 | 87.22 | 78.77 | 71.63 | 82.06 |
| | CsePL | **85.65**$^*$ | **66.66**$^\dagger$ | **76.94**$^\dagger$ | 95.09$^-$ | **83.14**$^-$ | **88.66**$^*$ | **80.28**$^\dagger$ | **71.99**$^-$ | **85.68**$^\dagger$ |
| 75% | DTC | 74.44 | 44.68 | 57.16 | 89.19 | 67.15 | 77.65 | 63.05 | 53.83 | 71.04 |
| | CDAC+ | 69.54 | 37.78 | 51.07 | 85.96 | 55.17 | 67.77 | 58.23 | 40.95 | 64.57 |
| | DeepAligned | 79.39 | 53.09 | 64.63 | 93.92 | 79.94 | 86.79 | 73.28 | 60.09 | 77.97 |
| | DCSC | 84.65 | 64.55 | 75.18 | 95.28 | 84.41 | 89.70 | - | - | - |
| | MTP-CLNN | 86.19 | 66.98 | 77.22 | 95.45 | 84.30 | 89.46 | 77.12 | 69.36 | 82.90 |
| | USNID | 87.41 | 69.54 | 78.36 | 96.42 | 86.77 | 90.36 | 80.13 | 74.90 | 85.66 |
| | CsePL | **87.70**$^-$ | **71.36**$^*$ | **81.93**$^\dagger$ | **96.58**$^-$ | **88.88**$^\dagger$ | **93.46**$^\dagger$ | **82.81**$^\dagger$ | **75.99**$^*$ | **87.80**$^\dagger$ |

Table 1: Main performance results on the new intent discovery across three public datasets. $\dagger$ denotes p-value<0.01, * denotes p-value<0.05 and - denotes p-value>0.05 under t-test.

dataset and contains 22,500 samples with 150 intents across 10 domains. **StackOverflow** (Xu et al., 2015) dataset is collected from Kaggle.com, which includes 20,000 samples over 20 intents.

In the experiments, we retain the same division of Banking77, Clinc150, and StackOverflow as delineated in Zhang et al. (2023). More experimental details can be found in Appendix A.2.

## 4.2 Evaluation Metrics

We adopt three commonly used metrics to evaluate the clustering performance: Normalized Mutual Information (NMI), Adjusted Rand Index (ARI) and Accuracy (ACC). To evaluate ACC, we use the Hungarian algorithm (Kuhn, 1955) to construct the mapping between predicted clusters and ground-truth intent categories. Note that ACC is the most important evaluation metric in our experiments.

## 4.3 Baselines

We extensively compare the proposed CsePL method with several SOTA baselines including **DTC** (Han et al., 2019), **CDAC+** (Lin et al., 2020), **DeepAligned** (Zhang et al., 2021c), **DCSC** (Wei et al., 2022), **MTP-CLNN** (Zhang et al., 2022) and the most recent **USNID** (Zhang et al., 2023). We leave the details of the baselines in Appendix A.1.

## 4.4 Main Results

### 4.4.1 NID Performance Comparison

We present the main performance comparison results in Table 1, where the best results are highlighted in **bold**. Generally speaking, our proposed CsePL achieves significant improvements compared with the previous baselines. Here, we present the result analyses from the following aspects:

**Our proposed CsePL learns cluster-friendly semantics for discovering new intents**: It can be seen that the proposed CsePL outperforms the baselines such as USNID and MTP-CLNN significantly and achieves new SOTA performances on three NID datasets. For example, compared with the USNID, our CsePL improves the ACC by 3.57%, ARI by 1.82%, and NMI by 0.29% on the Banking77 dataset with 75% known intent rate. It is worth mentioning that the Banking77 dataset is a fine-grained dataset collected from banking dialogues, whose utterances and intent labels contain rich semantic knowledge. This demonstrates that the CsePL can mine valuable and cluster-friendly semantic knowledge from the training data to enhance new intent clustering.

Interestingly, we find that although the proposed CsePL tends to perform worse than the strongest baseline USNID on the NMI metric for the 25%-StackOverflow setting, the results are generally

| Intent Type | Intent Label | Related Tokens |
|---|---|---|
| $\mathcal{C}_k$ | lost or stolen phone | or on lost charge statement extra stolen **supported** phone |
| | atm support | **supported** and cards currencies cute surfer schooner globe |
| | top up failed | failed extra on charge statement up top tornadoes hundreds |
| | supported cards and currencies | **cash** withdrawal declined **money** metropolitan why pickup desk |
| $\mathcal{C}_u$ | card not working | order physical receiving money withdrawal card clay |
| | exchange charge | exchange via app club cash crank pilot reset hit |
| | top up reverted | top up failed limits not sometimes entire get upper |
| | wrong amount of cash received | **receiving** transfer order physical losing lily **money** lost clay |

Table 2: Intent label suggestions: the related tokens that appear in the intent label are marked in red, while the related tokens that have similar meaning to the intent label tokens are highlighted in **bold**.

less stable as evidenced by the large p-value. We find that given different random seeds, CsePL would perform better or worse than the USNID on the 25%-StackOverflow. This might due to the difference on the set of known intents selected.

**Our proposed CsePL reduces in-domain overfitting for NID**: We can observe that our CsePL maintains its superior performance when confronting the predominance of a larger scale labeled data. For example, for the Clinc150 dataset with a 75% known intent rate setting, the CsePL surpasses the best-performing baseline USNID by margins of 3.1% in ACC, 2.11% in ARI, and 0.16% in NMI. It is worth noting that the Clinc150 dataset encompasses 150 distinct intents and possesses a greater number of labeled training utterances for known intents. This shows the efficacy of the CsePL in effectively mitigating in-domain over-fitting.

**Effect of different known intent rates**: From the Table 1, we can observe that the performances of all NID methods gradually decrease with the known intent rate going down. As the lower the known intent rate is, the less labeled data is available for guiding the model training, which leads to more difficult prior knowledge transfer for discovering new intents. However, with the decrease of the known intent rate, our proposed CsePL achieves more substantial improvements. For example, on the Banking77 dataset with a 25% known intent rate, the CsePL achieves 5.21% ACC improvement compared with the USNID, while the ACC improvements are 3.67% and 3.57% with known intent rates of 50% and 75%, respectively. These results further indicates that our proposed CsePL is more generalized in NID.

### 4.4.2 Suggesting New Intent Labels

In order to demonstrate that our proposed CsePL can solicit semantic knowledge from existing utterances and intent labels to suggest new intent cluster labels, we select four known intents and unknown intents from $\mathcal{C}_k$ and $\mathcal{C}_u$ respectively in the Banking77 dataset with the 25% known intent rate, and utilize their intent cluster representations learned in the ICRL stage to search for the most related tokens from the whole BERT-based vocabulary. We use the cosine similarity for the ranking and only present the most relevant tokens after filtering special tokens such as "[MASK]" and "[CLS]".

As reported in Table 2, for the known intents, the intent cluster representations learned by our CsePL can exactly retrieve the label tokens or semantic similar tokens from the BERT-based vocabulary. For example, given an intent *top up failed*, all tokens appearing in the intent label are retrieved by the intent cluster representation as the most related tokens. For the intent *supported cards and currencies*, our proposed CsePL can search for and distinguish the semantic similar tokens such as *cash* and *money* as the relevant tokens. This suggests that the intent cluster representations derived by our CsePL can accurately capture the semantics associated with their respective intent labels.

It is noteworthy that the CsePL is also capable of providing meaningful label suggestions for unknown intents. For instance, our CsePL picks up the token *exchange* from the vocabulary as the most relevant token for the intent *exchange char* in $\mathcal{C}_u$. Even for the unknown intent *wrong amount of cash received* with more complex semantics, the semantic similar tokens *receiving* and *money* are retrieved as the related tokens by the CsePL. This shows the ability of the CsePL in suggesting

| Methods | Banking77 | | |
|---|---|---|---|
| | NMI | ARI | ACC |
| 25% MTP-CLNN | 74.12 | 43.87 | 56.67 |
| USNID | 74.52 | 44.46 | 56.12 |
| CsePL | **76.31** | **47.39** | **59.17** |

Table 3: Results of early detection of new intents. For each unknown intent, only 20 utterances are available.

meaningful unknown intent labels.

### 4.4.3 Early Detection of New Intents

Early detection is a critical requirement for new intent discovery methods. To demonstrate this, we compare the performances of different methods when only a few utterances are available for each unknown intent. The results are reported in Table 3. We can observe that when the utterances for each unknown intent are limited, all methods perform worse than before, but the proposed CsePL significantly outperforms other two methods. It indicates the ability of the CsePL in discriminating new intents in the early stage. This also signals the importance of leveraging semantic knowledge.

### 4.5 Detailed Analysis

In this subsection, we conduct detailed analysis to explore the impact of each key component in the CsePL: 1) CsePL w/o PID: we entirely remove the PID stage and only use the model trained in the ICRL stage for NID. 2) CsePL w/o SemanticAlign: we remove the label semantic alignment (Equation 4) in the ICRL stage during training. 3) We also analyze the effect of predicted cluster number $K$. Note that we present results exclusively for the Banking77. While other datasets exhibit similar patterns but we omit them due to space limitation.

### 4.5.1 Effect of Prompt Discrimination

As shown in Table 4, we can observe that the performance of the CsePL substantially diminishes across all the known intent rates after removing the PID stage. Especially, when the known intent rate is set to 25%, the ACC of the CsePL considerably drops 9%+ while the NMI and ARI drop 5%+ and 10%+, respectively. It indicates that prompting the input with all the intent cluster representations learned during the ICRL stage can assist discriminating new intents. To delve deeper into the effectiveness of the PID, we also conduct experiments

| Methods | Banking77 | | |
|---|---|---|---|
| | NMI | ARI | ACC |
| 25% CsePL | **83.32** | **60.36** | **71.06** |
| - w/o PID | 77.88 | 50.21 | 61.38 |
| - w/o SemanticAlign | 82.12 | 58.09 | 68.75 |
| 50% CsePL | **85.65** | **66.66** | **76.94** |
| - w/o PID | 83.91 | 62.63 | 73.46 |
| - w/o SemanticAlign | 85.38 | 65.88 | 76.38 |
| 75% CsePL | **87.70** | **71.36** | **81.93** |
| - w/o PID | 86.91 | 69.23 | 79.40 |
| - w/o SemanticAlign | 87.39 | 70.84 | 81.24 |

Table 4: Ablation results on Banking77 dataset.

| Cluster Num | Methods | Banking77 | | |
|---|---|---|---|---|
| | | NMI | ARI | ACC |
| $K = 77$ (gold) | USNID | 81.94 | 56.53 | 65.85 |
| | CsePL | 83.32 | 60.36 | 71.06 |
| $K = 74$ (predicted) | USNID | 78.11 | 49.18 | 60.72 |
| | CsePL | 81.30 | 56.70 | 69.75 |

Table 5: Effect of estimating cluster number $K$.

that investigate various soft prompt initialization techniques during the PID stage. We leave the details of the experimental results in Appendix A.3.

### 4.5.2 Effect of Semantic Alignment

We also compare the model performance of removing semantic alignment process in the ICRL stage with the standard CsePL to explore the contribution of the semantic alignment. We find that eliminating the semantic alignment for intent cluster representation learning degrades the performance for new intent discovery. For example, the ACC drops from 71.06% to 68.75% on the Banking77 dataset with the 25% known intent rate without utilizing the semantics. This demonstrates the importance of the semantic alignment process.

### 4.5.3 Effect of Estimating Cluster Number $K$

We have been assuming the cluster number $K$ as a given hyper-parameter in the same fashion as baselines. However, in the practical dialogue systems, the number of clusters is unknown and it is important to predict $K$ for new intent discovery. Following the work of Zhang et al. (2021c), we predict the cluster number $K$ via an estimation algorithm. More details can be found in Appendix A.4. We present the model performances of different cluster number $K$ in Table 5. We can observe that although the performances of both

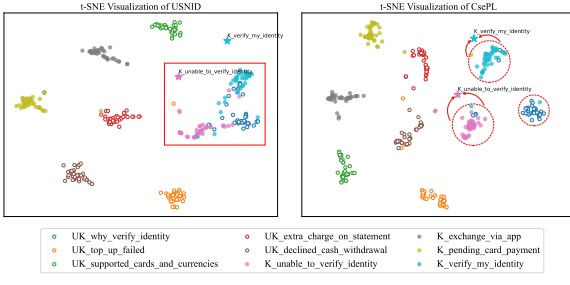

Figure 3: T-SNE visualization. The prefix "*UK_*" and "*K_*" denote unknown intents (hollow circles) and known intents (solid circles) respectively.

our CsePL and the SOTA baseline USNID decline with an inaccurate cluster number $K$, the proposed CsePL still achieves significant performance improvements over the USNID method. It shows that the proposed CsePL is more robust regarding the estimated cluster number $K$.

### 4.6 NID Representation Visualisation

In order to more intuitively analyze the effect of the proposed CsePL in representation learning, we present t-SNE visualizations comparing the leading baseline USNID and our CsePL approach, as illustrated in Figure 3. The USNID visualization reveals that data points of the unknown intent are distorted and dispersed within two known intent clusters *unable to verify identity* and *verify my identity*. This results in the in-domain over-fitting problem. Furthermore, this dispersion undermines the meaning of the newly learned intent cluster, as it encompasses instances from three distinct intents. Conversely, the visualization for the CsePL demonstrates how the label semantic alignment effectively aligns the intent cluster representations with the semantics of their corresponding labels. This process renders the unknown intent cluster, *why verify identity*, more coherent and less distorted. Additionally, with reduced noise in this cluster, its meaning becomes more discernible.

### 4.7 Error Analysis

In this subsection, we conduct an error analysis to delve into the problem of in-domain over-fitting and to evaluate the effectiveness of our proposed CsePL method. In Table 6, we present the ratio of unknown intent samples that the model wrongly classified as known intents. Additionally, we highlight the percentage of utterances that originate

| | Methods | UPK | KPE |
|---|---|---|---|
| 25% | USNID | 0.091 | 0.046 |
| | CsePL | 0.065 (*-28.57%*) | 0.041 (*-10.87%*) |

Table 6: Ratio of wrongly predicted intents. UPK denotes utterances that belong to **U**nknown intents but are inaccurately **P**redicted as **K**nown intents. KPE represents utterances that originate from **K**nown intents but are **P**redicted **E**rroneously.

from known intents but were inaccurately predicted. We can observe that the leading baseline USNID incorrectly classifies 9.1% of utterances with unknown intents as known intents. This misclassification rate is nearly twice that of known intents being predicted inaccurately, which stands at 4.6%. This implies that the presence of known intent data could excessively sway the clustering procedure, leading to the in-domain over-fitting problem. Compared with the USNID, the clustering outcomes derived from the CsePL demonstrate a diminished influence of the known intent data, leading to a notable reduction in both ratios.

## 5 Conclusion

In this paper, we reemphasized the importance of semantic knowledge in new intent discovery and proposed a Cluster semantic enhanced Prompt Learning (CsePL) method. Specifically, we designed two-level contrastive learning with label semantic alignment for intent cluster representation learning, and a soft prompting method to leverage the learned intent cluster representation for NID. Experimental results on three public datasets demonstrate the effectiveness of the CsePL. Extensive analyses further show that the CsePL not only significantly outperforms the existing baselines, but also implies new intent labels and detects the appearance of the new intents at an early stage.

### Acknowledgement

This research is supported by the Ministry of Education, Singapore, under its AcRF Tier 2 Funding (Proposal ID: T2EP20123-0052). Any opinions, findings and conclusions or recommendations expressed in this material are those of the author(s) and do not reflect the views of the Ministry of Education, Singapore.

## Limitations

We discuss the limitations from the following perspectives: (1) **Usage of LLMs**. Recently, large language models (LLMs) such as ChatGPT or GPT-4 have exhibited their outstanding performance on various NLP task, showing its abundance in semantic knowledge. Though BERT has advantage in relatively low resource consumption, we will look into how to leverage LLMs's knowledge for better NID. (2) **New intent labels**. Although our method has show the potential in suggesting new intent labels, we plan to further investigate the possibility of generating the whole label directly, which will be more useful. (3) **Early detection**. It is critical in deployed systems. We plan to further look into this aspect and conduct comprehensive experiments to test the limit on how early it can work.

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

# A Appendix

## A.1 Additional Details of Baseline Methods

In this work, we compare the proposed CsePL framework against several representative baselines including:

- **DTC** (Han et al., 2019): a semi-supervised deep embedding clustering method with a mechanism to estimate new intent numbers.
- **CDAC+** (Lin et al., 2020): a pseudo-labeling method the uses pairwise similarities to guide the clustering process.
- **DeepAligned** (Zhang et al., 2021c): an improved DeepClustering (Caron et al., 2018) that uses an alignment strategy to alleviate the label inconsistency problem.
- **DCSC** (Wei et al., 2022): a two-stage NID method with the dual-task process, which regards the soft pseudo cluster assignments as the supervision signals to train the model.
- **MTP-CLNN** (Zhang et al., 2022): a method that applies multi-task pre-training and nearest neighbors contrastive learning for NID.
- **USNID** (Zhang et al., 2023): a two-stage framework for both unsupervised and semi-supervised NID with an efficient centroid-guided clustering mechanism.

For fair comparison, the BERT-based model is used as the backbone in all aforementioned baselines. The external labeled dataset used in MTP-CLNN is also removed.

## A.2 Experimental Details

In the experiments, we keep the same split of Banking77, Clinc150 and StackOverflow as in (Zhang et al., 2023). For each dataset, we randomly select a certain ratio (25%, 50%, 75%) of intents as known intents, and sample 10% labeled instances from known intent categories to form a labeled subset, while the remaining instances are treated as the unlabeled data.

For the proposed CsePL, we adopt the BERT-model (Devlin et al., 2019) (*bert-base-uncased*, with 12 transformer layers) as the backbone model. The AdamW optimizer is used to update the model parameters. In the ICRL stage, we pre-train the model 100 epochs with a 20-epochs patience to validate early-stopping on the developments. The learning rate is set to $5 \times 10^{-5}$. To calculate

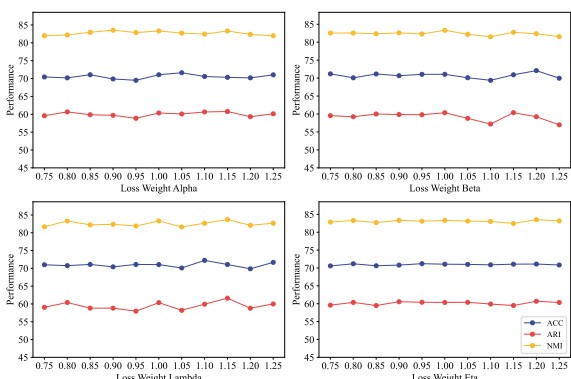

Figure 4: NID performance of distinct loss weights on the Banking77 dataset with 25% known intent rate.

contrastive loss, we employ a projection layer to transform the output representations of the backbone into a 768-dimensional vector space. The contrastive learning temperatures $\{\tau_1, \tau_2, \tau_3\}$ are uniformly set to 0.07. The values of $\{\alpha, \beta, \lambda, \eta\}$ are set to 1.0 to balance distinct losses in $\mathcal{L}_{ICRL}$. We also conduct extensive hyper-parameter exploration experiments to determine appropriate values for $\{\alpha, \beta, \lambda, \eta\}$. The detailed experimental results are reported in Figure 4. It's worth mentioning that our approach exhibits relative stability concerning these hyper-parameters. Thus, we select 1.0 as our definitive setting in our experiments.

In the PID stage, we train the model for 50 epochs with the learning of $2 \times 10^{-5}$. The output representations in this stage are mapped to a 128-dimensional vector space for contrastive learning. The temperature $\tau_4$ is set to 0.07 similarly. Note that all experimental results are averaged over 4 different random seeds.

## A.3 Results of Different Soft Prompt Initialization Methods

To further study the effect of the soft prompts in the PID stage, we compare the NID performance of different CsePL variants across various datasets, as illustrated in Table 7. Specifically, we incorporate two types of prompts: soft prompts initialized randomly and hand-crafted discrete prompts, and identically integrate them with input utterances in the PID stage for comprehensive experiments. We can observe that, when utilizing hand-crafted discrete prompts in the PID stage, the NID performance consistently declines across all datasets in comparison to the standard CsePL. The CsePL-*random*

| Known Intent Rate | Methods | Banking77 | | | Clinc150 | | | StackOverflow | | |
|---|---|---|---|---|---|---|---|---|---|---|
| | | NMI | ARI | ACC | NMI | ARI | ACC | NMI | ARI | ACC |
| 25% | CsePL | **83.32** | **60.36** | **71.06** | **94.07** | **79.65** | **86.16** | **74.88** | **64.92** | **79.47** |
| | CsePL-*random* | 82.13 | 58.22 | 69.45 | 93.36 | 77.23 | 84.74 | 74.27 | 63.57 | 77.25 |
| | CsePL-*manual* | 81.24 | 56.89 | 69.11 | 93.36 | 77.06 | 83.71 | 73.43 | 61.48 | 76.30 |
| 50% | CsePL | **85.65** | **66.66** | **76.94** | **95.09** | **83.14** | **88.66** | **80.28** | **71.99** | **85.68** |
| | CsePL-*random* | 85.45 | 65.50 | 75.91 | 94.63 | 81.41 | 87.85 | 78.60 | 70.31 | 84.20 |
| | CsePL-*manual* | 84.26 | 63.76 | 75.47 | 94.32 | 80.07 | 86.98 | 77.60 | 69.03 | 83.95 |
| 75% | CsePL | **87.70** | **71.36** | **81.93** | **96.58** | **88.88** | **93.46** | **82.81** | **75.99** | **87.80** |
| | CsePL-*random* | 86.80 | 69.57 | 79.97 | 95.87 | 85.98 | 91.02 | 79.96 | 72.16 | 85.55 |
| | CsePL-*manual* | 86.49 | 68.94 | 79.76 | 95.61 | 85.26 | 90.54 | 80.39 | 71.82 | 84.75 |

Table 7: NID performance comparison across different soft prompt initialization methods. CsePL-*random* and CsePL-*manual* represent the CsePL variants wherein the prompts in the PID stage are randomly initialized soft prompts and hand-crafted discrete prompts respectively.

employs soft prompts with random initialization in the PID stage and exhibits a slight performance uplift compared with the CsePL-*manual*. Yet, both of CsePL-*random* and CsePL-*manual* consistently lag behind the CsePL in the discovering new intents. This indicates that utilizing soft prompts, which are initialized based on the learned intent cluster representations, can provide the model with a comprehensive array of all intent options and effectively directs the model in matching input utterances to their respective intents.

## A.4 Estimate Cluster Number $K$

In practical dialogue systems, new intents emerge constantly and we cannot know the exact number of the intent clusters. In this paper, following the work of Zhang et al. (2021c), we take the fully usage of the well-initialized intent features to automatically estimate the intent cluster number $K$. Specifically, we first assign a big $K'$ as the initial intent cluster number. Then we directly use the well-pretrained model to extract the feature representations for the training data and perform the $K$-means algorithm to group these feature representations into different clusters. From these clusters, we can distinguish the dense and boundary-clear clusters as the real intent clusters, while the remaining low size clusters are filtered out. The filtering function can be formulated as follows:

$$K = \sum_{i=1}^{K'} \delta(|S_i| > t) \qquad (5)$$

where $|S_i|$ is the size the $i_{th}$ grouped cluster, $t$ is the threshold of filtering. $\delta(\cdot)$ is the indicator function, whose output is 1 if the condition is satisfied.

