# OpenReview forum: "ClusterPrompt: Cluster Semantic Enhanced Prompt Learning for New Intent Discovery"
_EMNLP/2023/Conference — EMNLP 2023 Findings_

### Official Review · Reviewer_ASNm · 2023-07-26

**Soundness:** 3

**Excitement:**

3: Ambivalent: It has merits (e.g., it reports state-of-the-art results, the idea is nice), but there are key weaknesses (e.g., it describes incremental work), and it can significantly benefit from another round of revision. However, I won't object to accepting it if my co-reviewers champion it.

**Paper Topic And Main Contributions:**

The paper proposes a semi-supervised method for new intent discovery in user utterances.
The authors of this paper recognize that previous works suffer from in-domain over-fitting and generate low-quality new intent clusters.
Hence, the paper present CsePL, which leverages two-level contrastive learning with label semantic alignment to learn representations of intent clusters.

**Reasons To Accept:**

1. Authors conducted extensive experiments to show the effectiveness of the proposed methods.
2. The improvements compared to baselines are not trivial.
3. The paper presents a detailed introduction to methods that makes readers easy to follow their ideas.

**Reasons To Reject:**

1. The motivation of this paper is not clear. The authors claim that previous studies paid less attention to the usage of semantics and the limitations described from line 71 to line 76 are vague and less intuitive. It's also unclear the relation between the proposed methods and these limitations.
2. Method in this paper contains multiple components. It's hard to determine the effectiveness of these proposed parts.
3. It's unclear what the augmented version is described in line 283.
4. The effectiveness of the sequence of soft prompts is unclear. My understanding is that all cluster representations are inserted into all data instances, so it's identical to all examples. Why can this help distinguish different clusters?
5. In PID, why do you use a pre-trained backbone to get token representations? The backbone is trained with the manual prompt but in the PID stage, utterances do not use prompts. I think this inconsistency would cause suboptimal representations for tokens.


**Reproducibility:**

3: Could reproduce the results with some difficulty. The settings of parameters are underspecified or subjectively determined; the training/evaluation data are not widely available.

**Reviewer Confidence:**

3: Pretty sure, but there's a chance I missed something. Although I have a good feel for this area in general, I did not carefully check the paper's details, e.g., the math, experimental design, or novelty.

---

> ### Author Rebuttal · Authors · 2023-08-29
>
> Sincerely thank you for your time and detailed comments on this work. Here are our answers to the questions.
>
> **Q1: The motivation of this paper is not clear. The authors claim that previous studies paid less attention to the usage of semantics and the limitations described from line 71 to line 76 are vague and less intuitive. It's also unclear the relation between the proposed methods and these limitations.**
>
> **A1:** We apologize for not making the motivation clear enough. To concisely describe our motivation, we add a data visualization for the strongest baseline USNID and our method.
>
> In the figure, the prefix "UK_" and "K_" refer to unknown intents (hollow circles) and known intents (solid circles) respectively. The solid stars indicate the positions of label representations for label semantic alignment.
>
> ![representation_visualization_graph]( https://anonymous.4open.science/r/emnlp2023-3858/representation_visualization.png )
>
>
> The limitations of existing works include two main aspects: "in-domain over-fitting" problem and "meaningless new intents" problem. For example, as illustrated in the data visualization of the strongest baseline USNID (left hand side), some data points of the unknown intent "why_verify_identity" have become distorted and scattered within other two known intent clusters: "unable_to_verify_identity" and "verify_my_identity". This suggests that the known intent data can significantly impact the distribution of unknown intent data, leading to the emergence of the "in-domain over-fitting" issue. It will also result in the learned new cluster having less meaningful semantics, as it comprises data points from three different intents.
>
> To alleviate these limitations, we proposed to train the model using label semantic alignment and the PID process. As the label semantics is more stable, we can observe from the data visualization of the CsePL (right hand side) that the label representations (solid stars) can attract their associated data points, forming compact and well-defined cluster regions. Hence, the "in-domain over-fitting" problem can be mitigated. Also, the meaning of the newly learned cluster would be more clear due to fewer noisy data points in this cluster.
>
> We will modify our illustration figure (Figure 1) in Introduction, modify the Method section accordingly and add these experiments to highlight our motivation.
>
> **Q2: Method in this paper contains multiple components. It's hard to determine the effectiveness of these proposed parts.**
>
> **A2:** Most of the components used in our method come from existing baselines, which have been already validated as effective, such as "utterance-level contrastive loss", "Cluster level contrastive loss" etc. To alleviate the "in-domain over-fitting" and "meaningless new intents" problems, we specifically propose two primary parts: label semantic alignment and the PID process. The label semantic alignment aims to align intent cluster representations to their corresponding label in the semantic space. For example, removing this part from our model can cause a performance drop in ACC from 71.06% to 68.75% in the Banking77 dataset with 25% known intent rate. The goal of the PID process is to provide the model with a comprehensive set of all intent options, assisting the model to match input utterances to their corresponding intents. For example, without the PID process, the ACC performance of the model drops from 71.06% to 61.38% under the setting of Banking77 dataset with 25% known intent rate. Detailed ablation results can be found in Table 4 in our manuscript.
>
> **Q3: It's unclear what the augmented version is described in line 283.**
>
> **A3:** Similar to [1], we use dropout to construct augmented examples for contrastive learning. Here, the augmented version is the dropout-augmented utterance representation of the input utterance.
> \
> [1]. Tianyu Gao, Xingcheng Yao, and Danqi Chen. 2021. Simcse: Simple contrastive learning of sentence embeddings. ArXiv, abs/2104.08821.
>
> **Q4: The effectiveness of the sequence of soft prompts is unclear. My understanding is that all cluster representations are inserted into all data instances, so it's identical to all examples. Why can this help distinguish different clusters?**
>
> **A4:**
> Thanks for the insightful comment. Within our method, intent cluster representations are trained to align their corresponding label representations via label semantic alignment. Thus, they can reflect the global intent distribution in the feature space. Utilizing them as soft prompt initializations provides the model with a comprehensive array of all intent options, effectively guiding the model to match input utterances to their respective intents.
>
> On the contrary, we randomly initialize the soft prompts and identically integrate them with all input utterances to conduct additional experiments. The experimental results are reported in Table 1. We can notice that the model has a performance decline on all three datasets without initializing the soft prompts with the aligned intent cluster representations. We will include these findings in the revised version of the manuscript.
>
> ```
> Table 1. Comparison of model performance using various soft prompt initializations.
> ```
> | Banking77         |              |         |        |         |
> | :-----:           | :-----:      | :-----: | :----: | :----:  |
> | Known Intent Rate | Method       | MNI     | ARI    | ACC     |
> | 0.25              | CsePL        | 83.32   | 60.36  | 71.06   |
> | 0.25              | CsePL-random | 82.13   | 58.22  | 69.45   |
> | 0.5               | CsePL        | 85.65   | 66.66  | 76.94   |
> | 0.5               | CsePL-random | 85.45   | 65.50  | 75.91   |
> | 0.75              | CsePL        | 87.70   | 71.36  | 81.93   |
> | 0.75              | CsePL-random | 86.80   | 69.57  | 79.97   |
>
> | Clinc150          |              |         |        |         |
> | :-----:           | :-----:      | :-----: | :----: | :----:  |
> | Known Intent Rate | Method       | MNI     | ARI    | ACC     |
> | 0.25              | CsePL        | 94.07   | 79.65  | 86.16   |
> | 0.25              | CsePL-random | 93.36   | 77.23  | 84.74   |
> | 0.5               | CsePL        | 95.09   | 83.14  | 88.66   |
> | 0.5               | CsePL-random | 94.63   | 81.41  | 87.85   |
> | 0.75              | CsePL        | 96.58   | 88.88  | 93.46   |
> | 0.75              | CsePL-random | 95.87   | 85.98  | 91.02   |
>
> | StackOverflow     |              |         |        |         |
> | :-----:           | :-----:      | :-----: | :----: | :----:  |
> | Known Intent Rate | Method       | MNI     | ARI    | ACC     |
> | 0.25              | CsePL        | 74.88   | 64.92  | 79.47   |
> | 0.25              | CsePL-random | 74.27   | 63.57  | 77.25   |
> | 0.5               | CsePL        | 80.28   | 71.99  | 85.68   |
> | 0.5               | CsePL-random | 78.60   | 70.31  | 84.20   |
> | 0.75              | CsePL        | 82.81   | 75.99  | 87.80   |
> | 0.75              | CsePL-random | 79.96   | 72.16  | 85.55   |
>
> **Q5: In PID, why do you use a pre-trained backbone to get token representations? The backbone is trained with the manual prompt but in the PID stage, utterances do not use prompts. I think this inconsistency would cause suboptimal representations for tokens.**
>
> **A5:** Thank you for your comment. The primary goal of our method is to first learn a global distribution of all intents, and then refine the model to better associate input utterances with their corresponding intents. Therefore, we believe that utilizing the backbone pre-trained in stage 1 as the feature extractor in PID would not cause the inconsistent problem. We conduct experiments on employing a new BERT-model as the feature extractor in PID, the model cluster performance drops dramatically. For example, under the setting of Banking77 dataset with 25% known intent rate, the ACC metric decreased from 71.06% to 41.4%.
>
> It should be noted that we intentionally employ different prompting methods in stage 1 and stage 2. In the first stage, we train the model with a manually constructed prompt to learn aligned intent cluster representations. In the second stage, we initialize the soft prompts with all the learned intent representations, integrating them with input utterances to further improve the model's performance. It provides the model with a comprehensive array of all intent options, effectively guiding the model to match input utterances to their respective intents.
>
> To verify whether different prompting methods in two stages cause the suboptimal issue, we conduct experiments by employing identical manual prompt in both two stages. As reported in Table 2, when using the identical manual prompt in both two stages, the model performance declines on all three datasets compared with the CsePL. We will include this analysis in the revised version to further support our arguments.
>
> ```
> Table 2. Comparison of model performance using soft prompts and identical manual prompt.
> ```
> | Banking77         |              |         |        |         |
> | :-----:           | :-----:      | :-----: | :----: | :----:  |
> | Known Intent Rate | Method       | MNI     | ARI    | ACC     |
> | 0.25              | CsePL        | 83.32   | 60.36  | 71.06   |
> | 0.25              | CsePL-manual | 81.24   | 56.89  | 69.11   |
> | 0.5               | CsePL        | 85.65   | 66.66  | 76.94   |
> | 0.5               | CsePL-manual | 84.26   | 63.76  | 75.47   |
> | 0.75              | CsePL        | 87.70   | 71.36  | 81.93   |
> | 0.75              | CsePL-manual | 86.49   | 68.94  | 79.76   |
>
> | Clinc150          |              |         |        |         |
> | :-----:           | :-----:      | :-----: | :----: | :----:  |
> | Known Intent Rate | Method       | MNI     | ARI    | ACC     |
> | 0.25              | CsePL        | 94.07   | 79.65  | 86.16   |
> | 0.25              | CsePL-manual | 93.36   | 77.06  | 83.71   |
> | 0.5               | CsePL        | 95.09   | 83.14  | 88.66   |
> | 0.5               | CsePL-manual | 94.32   | 80.07  | 86.98   |
> | 0.75              | CsePL        | 96.58   | 88.88  | 93.46   |
> | 0.75              | CsePL-manual | 95.61   | 85.26  | 90.54   |
>
> | StackOverflow     |              |         |        |         |
> | :-----:           | :-----:      | :-----: | :----: | :----:  |
> | Known Intent Rate | Method       | MNI     | ARI    | ACC     |
> | 0.25              | CsePL        | 74.88   | 64.92  | 79.47   |
> | 0.25              | CsePL-manual | 73.43   | 61.48  | 76.30   |
> | 0.5               | CsePL        | 80.28   | 71.99  | 85.68   |
> | 0.5               | CsePL-manual | 77.60   | 69.03  | 83.95   |
> | 0.75              | CsePL        | 82.81   | 75.99  | 87.80   |
> | 0.75              | CsePL-manual | 80.39   | 71.82  | 84.75   |

---

### Official Review · Reviewer_wx3r · 2023-08-04

**Typos Grammar Style And Presentation Improvements:** NA
**Soundness:** 3

**Excitement:**

4: Strong: This paper deepens the understanding of some phenomenon or lowers the barriers to an existing research direction.

**Missing References:**

NA

**Paper Topic And Main Contributions:**

This work is related with problem of discovering new intent categories from user utterances. Building upon the observation that existing approaches emphasize too much on relation among utterances and pay less attention to employing the semantics, this work presents Cluster Semantic Enhanced Prompt Learning (CsePL) for discovering new intents. The proposed approach leverages two-level contrastive learning with label semantic alignment to learn meaningful representations of intent clusters which are then utilized as soft prompt initializations for discriminating new intents, reducing the dominance of existing intents. Experimentations on three datasets (BANKING77, CLINC150 and StackOverflow) demonstrate that the proposed method achieves superior results compared to baselines.

**Questions For The Authors:**

See weakness section.

**Reasons To Accept:**

1. This papers tackles and important and relevant problem of discovering new intents from the existing ones. In order the enhance the capabilities of a dialogue system, new intent discovery is necessary.

2. A good number of baselines are studied and evaluated in order to establish the superiority of the proposed method.

3. Contrastive learning has been employed for capturing the semantics.

4. Several baselines has been employed for comparison.

5. Use of label semantic alignment is interesting.

6. Significant improvements are reported with insightful discussion.

**Reasons To Reject:**

1. The proposed method underperforms on NMI metric in comparison with USNID under some settings.

**Reproducibility:**

4: Could mostly reproduce the results, but there may be some variation because of sample variance or minor variations in their interpretation of the protocol or method.

**Reviewer Confidence:**

4: Quite sure. I tried to check the important points carefully. It's unlikely, though conceivable, that I missed something that should affect my ratings.

---

> ### Author Rebuttal · Authors · 2023-08-29
>
> Thank you so much for your recognition of our work, and we will improve our paper further.
>
> **Q1: The proposed method underperforms on NMI metric in comparison with USNID under some settings.**
>
> **A1:** We appreciate your observation. NMI is a metric used to measure the agreement between the ground truth clusters and the predicted clusters. In our experiments, ACC is the most important metric to evaluate the overall model performance. Our method outperforms the strongest baseline USNID by 1%~5% on ACC metric across three datasets. This indicates the effectiveness of the proposed method.
>
> Regarding the NMI metric, we find that due to the dataset characteristics, it is possible for the semantics of some short intent labels to overlap. For example, in the Clinc150 dataset, the semantics of the labels "order_status" and "order" overlap heavily. This occasionally leads to misclassifications of utterances between these two intents. We will further explore this limitation. One possible way is to enhance the short labels with domain knowledge such as using LLMs to rewrite the label with more details.

---

### Official Review · Reviewer_QMDg · 2023-08-07

**Typos Grammar Style And Presentation Improvements:** 1. The "related work" section overlap…
**Soundness:** 4

**Ethical Concerns:**

Yes

**Excitement:**

4: Strong: This paper deepens the understanding of some phenomenon or lowers the barriers to an existing research direction.

**Missing References:**

1. Caron, Mathilde, et al. "Deep clustering for unsupervised learning of visual features." Proceedings of the European conference on computer vision (ECCV). 2018.
2. Zhang, Hanlei, et al. "TEXTOIR: An Integrated and Visualized Platform for Text Open Intent Recognition." Proceedings of the 59th Annual Meeting of the Association for Computational Linguistics and the 11th International Joint Conference on Natural Language Processing: System Demonstrations. 2021.

**Paper Topic And Main Contributions:**

This paper focuses on the discovery of new intent categories from user utterances, emphasizing the importance of utilizing semantic evidence in the process.
The primary contributions include the introduction of Cluster Semantic Enhanced Prompt Learning (CsePL), which employs a two-level contrastive learning mechanism combined with label semantic alignment for effective intent cluster representation. The paper highlights how the CsePL method can suggest pertinent intent labels and enables early identification of new intents (when utterances of unknown intents are few).

**Questions For The Authors:**

1. The core innovation appears to lie in the application of label semantic alignment in ICRL and the use of soft prompts in PID. These aspects should be clearly highlighted to better delineate the unique contributions of this work.
2. About "in-domain over-fitting" and "meaningless new intent problems”, detailed evidence, possibly involving case studies, model comparisons or visual representations of the model performance, could strengthen these claims.
3. The total loss function in stage 1/2 needs to be summarized, including how to balance the losses.
4. The "meaningless new intent" problem is actually the "trivial solution" in traditional clustering, and related works should be investigated.

**Reasons To Accept:**

1. The proposed method reiterates the importance of semantics in new intent discovery, overcoming the limitations of previous methods that suffered from over-fitting and data distortion.
2. Extensive experiments on three public datasets proves the superiority of their method over existing ones but also the capability of suggesting meaningful intent labels.
3. The author conducted a commendable comparison with the latest state-of-the-art (SOTA) method, USNID, and carried out significant tests. This is highly praiseworthy.

**Reasons To Reject:**

1. The novelty of the paper is somewhat vague and needs to be more concise. The primary components seem to integrate effective elements of previous works, including SCL, Aligned, and CLNN, among others.
2. The authors' claims of solving "in-domain over-fitting" and "meaningless new intent problems" require more in-depth analysis and visualization.

**Reproducibility:**

4: Could mostly reproduce the results, but there may be some variation because of sample variance or minor variations in their interpretation of the protocol or method.

**Reviewer Confidence:**

4: Quite sure. I tried to check the important points carefully. It's unlikely, though conceivable, that I missed something that should affect my ratings.

---

> ### Author Rebuttal · Authors · 2023-08-29
>
> We sincerely thank you for your time and valuable comments. We answer the questions as below.
>
> **Q1: The core innovation appears to lie in the application of label semantic alignment in ICRL and the use of soft prompts in PID. These aspects should be clearly highlighted to better delineate the unique contributions of this work.**
>
> **A1:** We sincerely appreciate your suggestion. We will revise the manuscript as below to highlight our contribution on label semantic alignment and soft prompt in PID:
>
> - **Introduction**: Instead of showing an illustration figure (Figure 1) for the new intent discovery task, we will provide an illustrative figure to emphasize the effect of label semantic alignment.
> - **Methodology**: We will modify our model figure (Figure 2) to highlight the label semantic alignment of our method. For the detailed methodology section, we will specifically update Section 3.3 to emphasize the Semantic Alignment.
> - **Experiment**: We will add data visualization results and error analysis results to showcase the effect of label semantic alignment and soft prompts in PID. More details in Q2&A2.
>
> **Q2: About "in-domain over-fitting" and "meaningless new intent problems", detailed evidence, possibly involving case studies, model comparisons or visual representations of the model performance, could strengthen these claims.**
>
> **A2:** Thank you for your suggestion. To strength our claims of "in-domain over-fitting" and "meaningless new intents" problems, we add **(1) an error analysis** and **(2) a data visualization**, which will be incorporated to the Experiment Section.
>
> **(1) an error analysis**
>
> We report the ratio of unknown intent samples that the model wrongly predicted as known intents and the ratio of utterances belong to known intents but are predicted wrongly. In the table below, we notice that 9.1\% of unknown intent utterances are erroneously classified as known intents by our strongest baseline USNID. It almost doubles the ratio of known intents predicted wrongly. This observation suggests that the known intent data might overly influence the clustering process, thereby giving rise to the issue of "in-domain over-fitting." Compared with the baseline, the clustering results achieved by the CsePL exihbit a reduced dominance of the known intent data, resulting in a significant decrease in both ratios.
>
> ```other
> Table 1. Ratio of intents predicted wrongly.
> ```
>
> | **method** | **label unknown, predicted as known** | **label known, predicted error** |
> | ---------- | ------------------------------------- | -------------------------------- |
> | USNID      | 0.091                                 | 0.046                            |
> | CsePL      | 0.065 (**-28.57\%**)                   | 0.041 (**-10.87\%**)              |
>
> **(2) a data visualization**
>
> We display the data representation visualization for USNID and our CsePL. As depicted in the following figure, the prefix "UK_" and "K_" denote unknown intents (hollow circles) and known intents (solid circles) respectively. The solid stars are the locations of label representations for label semantic alignment. The visualization is shown as below:
>
> ![representation_visualization_graph]( https://anonymous.4open.science/r/emnlp2023-3858/representation_visualization.png )
>
> From the USNID baseline visualisation (left hand side), we observe that the data points of the unknown intent "why_verify_identity" are distorted and dispersed within two known intent clusters: "unable_to_verify_identity" and "verify_my_identity". This observation indicates that the labeled data exert a dominant influence on unknown intent data, leading to the "in-domain over-fitting" problem. It will also cause the meaning of the learned new cluster less meaningful, as it contains instances from three intents.
>
> For our CsePL method (right hand side), we observe that the label semantics represented as stars will attract their corresponding utterance representations. It helps to leave the unknown intent "why_verify_identity" cluster less distorted. Also, since less noise appears in this cluster, the meaning of the cluster would be more clear.
>
> **Q3: The total loss function in stage 1/2 needs to be summarized, including how to balance the losses.**
>
> **A3:** We apologize for the initial omission of the total loss functions and the weight analysis for these losses. The total loss functions are formulated as follows:
>
> ***In stage 1:***
>
> $$
> L_{stage1} = \alpha \cdot L_{ce} + \beta \cdot L_{utt}^{cl} + \gamma \cdot L_{utt}^{ucl} + \lambda \cdot L_{clu}^{cl} + \eta \cdot L_{align}^{cl}
> $$
>
> where $L_{utt}^{cl}$, $L_{clu}^{cl}$ and $L_{align}^{cl}$ are defined in Eq(1), Eq(2) and Eq(4) respectively. $L_{ce}$ is the cross-entropy loss and $L_{utt}^{ucl}$ is the unsupervised contrastive learning loss. In reality, the hyper-parameters $\{\alpha, \beta, \gamma, \lambda, \eta\}$ are set to 1.0. We determined the optimal hyper-parameters by conducting experiments on the development set. The table below illustrates the fluctuation in overall performance with variations in $\eta$ for the Banking77 dataset with 25\% known intent rate. We carefully considered a range of values for $\eta$, ranging from 0.75 to 1.25. We notice that any subtle adjustment to $\eta$ does not appear to yield significant impacts on the overall performance. This observation also demonstrates the robustness and stability of our method.
>
> ```other
> Table 2. Performance of the model with varying loss weights.
> ```
>
> | $\eta$ | **MNI**   | **ARI**   | **ACC**   |
> | -------- | --------- | --------- | --------- |
> | 0.75     | 82.86     | 59.60     | 70.58     |
> | 0.80     | 83.28     | 60.37     | 71.17     |
> | 0.85     | 82.69     | 59.50     | 70.62     |
> | 0.90     | 83.33     | 60.56     | 70.81     |
> | 0.95     | 83.10     | 60.42     | 71.20     |
> | **1.00** | **83.32** | **60.36** | **71.06** |
> | 1.05     | 83.13     | 60.40     | 71.01     |
> | 1.10     | 83.02     | 59.91     | 70.88     |
> | 1.15     | 82.46     | 59.50     | 71.07     |
> | 1.20     | 83.52     | 60.69     | 71.10     |
> | 1.25     | 83.18     | 60.35     | 70.84     |
>
> We observed similar trends with other hyper-parameters for the $L_{stage1}$, and we plan to supplement this analysis with other hyper-parameters in the revised manuscript.
>
> ***In stage 2:*** The loss function in this stage is composed of a single contrastive learning loss as below:
>
> $$
> L_{stage2} = L_{PID}
> $$
>
> where $L_{PID}$ is defined in line 372.
>
> **Q4: The "meaningless new intent" problem is actually the "trivial solution" in traditional clustering, and related works should be investigated.**
>
> **A4:** We are grateful for your suggestion. We will add a new section to explore the "trivial solution" problem in clustering and discuss how it connects to and differs from the "meaningless new intent" problem. Thanks for providing the related works. We will add discussion about these and other related works.

---

### Meta-Review · Area_Chair_Fqee · 2023-09-12

**Recommendation:** 3

**Metareview:**

This work is concerned with the challenge of discovering new intent categories from user utterances, and introduces Cluster Semantic Enhanced Prompt Learning (CsePL) as a means to discover new intents. The proposed approach employs a two-level contrastive learning framework, incorporating label semantic alignment to acquire meaningful representations of intent clusters. These representations are subsequently employed as soft prompt initializations for distinguishing new intents, thereby reducing the influence of pre-existing intent categories. Experimental evaluations demonstrate that the proposed method outperforms baseline approaches. While the novelty of the paper may require further clarification and conciseness, the primary components of the proposed approach appear to integrate elements from previous works.

---

### Decision · Program_Chairs · 2023-10-07

**Decision:**

Accept-Findings

**Comment:**

This work is concerned with the challenge of discovering new intent categories from user utterances, and introduces Cluster Semantic Enhanced Prompt Learning (CsePL) as a means to discover new intents. The proposed approach employs a two-level contrastive learning framework, incorporating label semantic alignment to acquire meaningful representations of intent clusters. These representations are subsequently employed as soft prompt initializations for distinguishing new intents, thereby reducing the influence of pre-existing intent categories. Experimental evaluations demonstrate that the proposed method outperforms baseline approaches. While the novelty of the paper may require further clarification and conciseness, the primary components of the proposed approach appear to integrate elements from previous works.